# ROS-Mediated Fragmentation Alters the Effects of Hyaluronan on Corneal Epithelial Wound Healing

**DOI:** 10.3390/biom13091385

**Published:** 2023-09-13

**Authors:** Xiao Lin, Isabel Y. Moreno, Lawrence Nguyen, Tarsis F. Gesteira, Vivien J. Coulson-Thomas

**Affiliations:** College of Optometry, University of Houston, 4401 Martin Luther King Boulevard, Houston, TX 77204-2020, USA; xiao.lynn5@gmail.com (X.L.); iymoreno@cougarnet.uh.edu (I.Y.M.); lnguye76@CougarNet.UH.EDU (L.N.); tgferrei@central.uh.edu (T.F.G.)

**Keywords:** high-molecular-weight hyaluronan, low-molecular-weight hyaluronan, oxidative stress, wound healing, corneal epithelium

## Abstract

A buildup of reactive oxygen species (ROS) occurs in virtually all pathological conditions. Hyaluronan (HA) is a major extracellular matrix component and is susceptible to oxidation by reactive oxygen species (ROS), yet the precise chemical structures of oxidized HA products (oxHA) and their physiological properties remain largely unknown. This study characterized the molecular weight (MW), structures, and physiological properties of oxHA. For this, high-molecular-weight HA (HMWHA) was oxidized using increasing molar ratios of hydrogen peroxide (H_2_O_2_) or hypochlorous acid (HOCl). ROS lead to the fragmentation of HA, with the oxHA products produced by HOCl exhibiting an altered chemical structure while those produced by H_2_O_2_ do not. HMWHA promotes the viability of human corneal epithelial cells (hTCEpi), while low MWHA (LMWHA), ultra-LMWHA (ULMWHA), and most forms of oxHA do not. HMWHA and LMWHA promote hTCEpi proliferation, while ULMWHA and all forms of oxHA do not. LMWHA and some forms of oxHA promote hTCEpi migration, while HMWHA does not. Finally, all native forms of HA and oxHA produced by HOCl promote in vivo corneal wound healing, while oxHA produced by H_2_O_2_ does not. Taken together, our results show that HA fragmentation by ROS can alter the physiological activity of HA by altering its MW and structure.

## 1. Introduction

Reactive oxygen species (ROS) are molecules, ions, or radicals that have a non-static bond or an unpaired valance shell of electrons and are therefore highly reactive, such as superoxide and nitric oxide, and non-radical ROS, such as hydrogen peroxide (H_2_O_2_) [1]. Normally, atoms have paired electrons that spin symmetrically; however, with unpaired electrons, the spinning movements tend to destroy the surrounding structures. Therefore, although low levels of ROS are necessary for tissue homeostasis [2,3], excess ROS are toxic to cells and may lead to apoptosis, necrosis, and extracellular matrix (ECM) degradation, thus damaging tissues [4,5,6]. ROS can be produced endogenously due to aerobic metabolism, hypoxia, and immune activation [7,8]. For instance, in the process of oxidative phosphorylation, superoxide is produced by mitochondrial respiratory complexes [7]. ROS can also be induced by exogenous factors, such as UV light and radiation [4]. The detoxification of ROS is achieved by antioxidant enzymes that specifically catalyze the reduction of different ROS to water and a less reactive chemical, or by nonenzymatic antioxidants, such as DL-alpha-tocopherol acetate (vitamin E), which act as free radical scavengers [9]. A delicate balance between ROS generation and antioxidant defense mechanisms is necessary to maintain normal physiological functions and homeostasis [8]. Unbalanced oxidative chemical reactions can result in a buildup of excess ROS and lead to oxidative stress [10]. Oxidative stress is implicated in various diseases ranging from osteoarthritis, diabetes, and cancer to common ocular pathologies [10,11,12,13,14,15,16]. Most, if not all, pathological conditions go hand in hand with oxidative stress and consequently generate toxic levels of ROS [9,10,13]. Moreover, oxidative stress is an accepted component of aging and senescence [6,17,18]. The pathological outcomes of oxidative stress are attributed to the ROS-mediated modification of vital macromolecules, including nucleosides, lipids, and glycosaminoglycans (GAGs) in organisms [10,19,20,21,22,23,24,25]. 

Hyaluronan (HA) is a non-sulfated GAG that is ubiquitous in vertebrate tissues and fluids [26]. In mammals, HA is synthesized by membrane-bound HA synthases (HASs) and extruded into the extracellular space to be assembled into the ECM [27]. In humans, synovial fluid and the umbilical cord are rich in HA, in which concentrations can be as high as ~4 mg/mL [26,28]. In blood, the concentration of HA ranges between 10 and 100 μg/mL [26]. In the eye, HA is abundant in the aqueous humor, vitreous humor, limbal stem cell niche, limbal anterior stroma, and conjunctiva [29,30]. In the cornea, HA plays an important role in limbal epithelial stem cell fate decisions, corneal wound healing, and lymphangiogenesis [30,31,32,33]. Following corneal epithelial injuries, HA expression increases and extends from the limbus into the peripheral cornea [30,33]. HA confers structural support to tissues and cells and also exerts biological functions via the interaction with HA receptors, triggering signaling cascades that can regulate the expression of the genes required for cell proliferation, migration, differentiation, and angiogenesis [34,35]. Substantial evidence is mounting showing that the physiological properties of HA are dependent on the molecular weight (MW) of HA [36]. In tissues, HA exists in primarily two forms, high-molecular-weight HA (HMWHA) and low-molecular-weight HA (LMWHA) [36]. HMWHA is primarily correlated with maintaining tissue homeostasis and morphogenesis [37], while LMWHA is usually correlated with pathogenesis and inflammation [38,39].

In tissues, HA has also been shown to have antioxidant properties by suffering oxidative damage in lieu of other compounds [40,41,42,43]. Particularly, structural characteristics, such as decreased sulfation and β-linked structural units, make certain glycosaminoglycans (GAGs) more susceptible to be oxidized. In this respect, HA is particularly susceptible since it is non-sulfated and contains β-linkages between the repeating units of β-1,4-d-glucuronic acid and β-1,3-N-acetyl-d-glucosamine [44]. Additionally, other GAGs have some resistance to ROS due to the presence of a covalently bound core protein at the reducing end, which is not the case with HA [20,21,22,23,24,25,45]. It has been established that HA is highly susceptible to be oxidized by ROS, resulting in the fragmentation of the HA chain [21,22,23,24,25,46,47,48]. Studies have also demonstrated that different ROS yield distinct HA products after fragmentation [23,46,49]. Additionally, HA intermediate radicals can be formed during a redox reaction, and several potential structural rearrangements have been proposed for HA after oxidation [50]. The precise chemical structure of the oxidized HA products has been shown to depend on the specific type of ROS that fragments the HA chain [21,22,23,24,25,46]. For example, HA is depolymerized by singlet oxygen, generating modified tertiary structures [46] and by ozonolysis, forming a free aldehyde group at the non-reducing end [49]. In the case of hypochlorous acid (HOCl), the production of HA-based intermediate structures, such as chloramides and amidyl radicals, has been demonstrated [21,22,23,24,25]. HOCl has been shown to react with N-acetylglucosamine residues, and in the presence of metal ions, a lactone is believed to be generated from the N-acetyl glucosamine unit [21]. Although the effects of ROS on HA have been the subject of various studies, to the best of our knowledge, no studies to date have investigated whether oxidized HA products (oxHA) exhibit altered physiological properties when compared to native HA. 

Corneal epithelial abrasions are the most common type of eye injury requiring urgent care in the United States [51]. Although most simple abrasions heal rapidly, more severe wounds can result in persistent inflammation, bacterial infection, scarring, and corneal perforation, which can result in visual impairment and blindness [52,53]. We have previously shown that HA is upregulated in the cornea after an injury and has a major role in regulating corneal wound healing and pathological corneal lymphangiogenesis, which can occur after an injury [32]. Furthermore, studies have demonstrated that ROS increase in the cornea after an injury, and, consequently, that oxHA is produced [2,15,54,55,56]. In this study, we investigated the potential effects of ROS-mediated HA fragmentation on corneal wound healing. For such, various forms of oxHA were generated by the controlled oxidation of HMWHA using increasing molar ratios of H_2_O_2_ or HOCl. The MW and chemical structure of oxHA were characterized using agarose gel electrophoresis, gel filtration high-pressure liquid chromatography (HPLC) and nuclear magnetic resonance (NMR). Thereafter, the effects of oxHA and size-matched native HA were evaluated on corneal wound healing in vitro by assaying cell behaviors associated with wound healing, namely, viability, proliferation, and migration, as well as in vivo using corneal debridement wounds in mice.

## 2. Materials and Methods

### 2.1. Fragmentation of HMWHA to Generate oxHA

HMWHA (Cat #01473, 2.67 × 10^6^ Da, Lifecore Biomedical, Chaska, MN, USA) was subjected to oxidation with either H_2_O_2_ or HOCl. H_2_O_2_ or HOCl solutions were freshly prepared by diluting commercially available H_2_O_2_ (Cat #216763, Sigma-Aldrich, St. Louis, MO, USA) and NaOCl (Cat #425044, Sigma-Aldrich; available as chlorine 10–15%) in degassed autoclaved deionized water. Prior to the oxidation reaction, the HA solution was prepared and left under gentle agitation for 24 h to allow for full solvation [57]. At pH of 7.4, the dissociation of NaOCl (pK_a_ = 7.53) leads to a nearly equimolar concentration of HOCl and ClO^-^ ions, which we hereafter refer to as HOCl. In water, H_2_O_2_ decomposes into H_2_O and O_2_, which do not absorb light in the spectrum of 200–400 nm; therefore, the concentration of H_2_O_2_ can be determined by measuring the concentration of peroxide, which has a positive linear relationship between its optical density and concentration [58]. The starting concentration of H_2_O_2_ and HOCl was determined by photometric methods, as previously described [58,59]. Briefly, H_2_O_2_ was diluted 1000 times, and the concentration was determined using the extinction coefficient (ε) of 43.6 M^−1^·cm^−1^ at 240 nm with a NanoDrop™ 2000 spectrophotometer [58,59]. For HOCl, the NaOCl solution was diluted 1000 times in 0.01 M taurine solution followed by the measurement of absorbance of the mono-chloramine derivative. The calculation was based on the assumption that the mono-chloramine derivative has an ε of 429 M^−1^·cm^−1^ at 252 nm [59]. H_2_O_2_ solutions were prepared at a concentration of 37.5 µM and 187.5 µM, and HOCl solutions were prepared at a concentration of 37.5 µM, 187.5 µM, 937.5 µM, and 4687.5 µM and immediately mixed with 0.2% (*w*/*v*) HA at different molar ratios for 4 h at 37 °C while being protected from light (Figure 1). oxHA was then purified from the reactions by gel filtration using PD10 Sephadex G-25 desalting columns (Cat #17085101, GE Healthcare Bio-Sciences AB, Uppsala, Sweden) [60,61], thereby removing any residual ROS, and single-use aliquots stored at −80 °C to minimize freeze and thaw cycles. HA controls were produced by incubating and processing HA standards of different molecular weights alongside the oxidized HA samples, namely HMWHA, Mid-range-HA, (Cat #GLR004, R&D Systems, Minneapolis, MN, USA), LMWHA (Cat #026564, Lifecore Biomedical, Chaska, MN, USA), and ultra-low-molecular-weight HA (ULMWHA, #GLR003, R&D Systems, Minneapolis, MN, USA), but without them being subjected to H_2_O_2_ or HOCl. ROS controls were produced by incubating and processing H_2_O_2_ and HOCl in the absence of HA. ROS controls had no effect on hTCEpi or on corneal wound healing, confirming that any remaining ROS were successfully removed from all oxHA samples by PD10.

### 2.2. HA Molecular Weight by Agarose Gel Electrophoresis

Agarose gel electrophoresis was used to estimate the molecular weight of oxHA, as previously shown [62,63]. For this, 7 μg of oxHA and HA standards, including HMWHA and LMWHA, were lyophilized using a centrifugal vacuum concentrator (Eppendorf Vacufuge), and each sample was resuspended in 10 μL of 10 M deionized formamide (Millipore Sigma, S4117, Burlington, MA, USA) and kept at 4 °C overnight. HA of known MWs, 500 kDa and 50 kDa (Echelon Biosciences, Salt Lake City, UT, USA), was used to prepare a HA ladder mix at 0.7 μg/μL. A 0.5 cm thick 1% agarose gel (Omnipur Agarose, 2125, Calbiochem, San Diego, CA, USA 2125) was prepared in 1× TAE (Biotech USA, A0033, Minneapolis, MN, USA) and pre-run for 6 h at a constant voltage of 80 V (OWL EasyCast™ systems) to remove any impurities. A total of 2 μL of 0.2% bromophenol blue loading solution was added to previously prepared oxHA samples, HA standards, and HA ladder for a final volume of 12 μL and loaded onto the gel thereafter. Electrophoresis was carried out at a constant voltage of 100 V until the bromophenol blue tracking dye migrated to approximately 75% of the length of the gel. The gel was then transferred into a dish and equilibrated with 30% ethanol in water with gentle rocking for 1 h at room temperature, followed by 6.25 µg/mL Stains-All solution (#H32127, Alfa Aesar, Haverhill, MA, USA) at room temperature, protected from light. The next day, the staining solution was discarded and replaced with water for an hour. The gel was de-stained by being exposed to light for 20–30 min and imaged using the Biorad imager with white light. 

### 2.3. HA Molecular Weight Measurement by Gel Filtration High-Pressure Liquid Chromatography

Size exclusion chromatography (SEC) was used to characterize the molecular weight of HA, as previously shown [64]. For such, two Advanced Bio SEC 300 Å 4.6 × 300 mm columns (Agilent) and one Zorbax GF-250 4.6 × 250 mm column (Agilent) were used in tandem with the UltiMate™ 3000 Dionex system (Thermo Fischer Scientific, Waltham, MA, USA). The mobile phase used was 0.1 M sodium phosphate buffer containing 20 μM EDTA at pH of 6.8 and a flow rate of 0.3 mL/min. Temperature was maintained at 35 °C using a Dionex Ultimate^TM^ 3000 RS column compartment (Thermo Scientific™), and a Dionex Ultimate^TM^ 3000 versatile UV wavelength detector (Thermo Scientific™) was used set at 204 nm and 280 nm.

### 2.4. Structural Analysis by NMR

^1^H-NMR was used to evaluate the oxHA products. ^1^H-NMR spectra were acquired at room temperature with a Bruker 600 Advance III NMR spectrometer using a D_2_O solvent, as previously shown [57]. For such, oxHA in the LMWHA range for H_2_O_2_ and oxHA in the ULMWHA range for HOCl and native ULMWHA at a volume of 0.8 mL were transferred to a 5 mm NMR tube with trimethylsilylpropanesulfonic acid (DSS) as an internal reference. The analysis was performed using the MestreNova software, version 10.0 (Mestrelab Research S.L.). Each spectrum was estimated in the range of 400 to 4000 cm^−1^, with an average of 200 scans.

### 2.5. Corneal Epithelial Cell Culture Conditions

A human telomerase-immortalized corneal epithelial cell line (hTCEpi) (kindly provided by Dr. James Jester, UC Irvine) [65,66] was used in this study. hTCEpi was maintained in serum-free keratinocyte culture medium (DermaLife^®^ Basal media kit; Lifeline Cell Technology, Frederick, MD, USA) supplemented with 6 mM L-Glutamine, 1.0 μM epinephrine, 5 μg/mL recombinant human TGF-α, 100 ng/mL hydrocortisone hemisuccinate, 5 μg/mL rh-insulin, 5 μg/mL apo-transferrin, 30 mg/mL gentamicin, 0.4% bovine pituitary extra, and 15 μg/mL amphotericin B at 37 °C in 5% CO_2_. The cells were sub-cultured when reaching 70–90% confluence with 0.25% trypsin-EDTA (Gibco Ref 25200-056 Lot# 1933273). For all experiments, cells were plated onto multi-well dishes in complete media to which oxHA, HMWHA, LMWHA, ULMWHA (all prepared in PBS), or PBS (vehicle control) were added. 

### 2.6. Cell Viability Assay

The effect of oxHA on the viability of hTCEpi was assessed using Cell Counting Kit-8 (CCK-8—APExBIO Technology LLC, Houston, TX, USA). In this assay, the viability of the cells was assessed by quantifying mitochondrial activity, which was accomplished by measuring the formation of a water-soluble formazan dye upon reduction of tetrazolium salt (WST-8) catalyzed by dehydrogenases in live cells. Cells were trypsinized and seeded in 96-well plates at a density of 3000 cells/well. Media only (blank control) and cells in the media (media control) were used as controls. After 44 h, 10 μL of the CCK-8 reagent was added to each well and incubated for 4 h at 37 °C. Finally, the formazan product was analyzed by reading absorbance at 450 nm using a spectrophotometer microplate reader (FLUOstar Omega; BMG Labtech). Each experiment was performed independently in triplicate and carried out three times.

### 2.7. Cell Proliferation Assay

The effect of oxHA on the proliferation of hTCEpi was determined using the BrdU Cell Proliferation Assay Kit (Cat #2750; EMD Millipore, MA, USA). This assay labels proliferating cells during cell division by the incorporation of BrdU, an analog for thymidine, during de novo DNA synthesis. Briefly, hTCEpi was trypsinized and seeded in 96-well plates at a density of 5000 cells/well in the presence of 0.02% (*w*/*v*) oxHA, HMWHA, LMWHA, ULMWHA (all prepared in PBS), or PBS (vehicle control). The cells were incubated at 37 °C and 5% CO_2_ for 24 h, after which BrdU was added to the media, and the cells were incubated for an additional 8 h. Media only (no cells) and cells without the addition of BrdU were both used as negative controls. Finally, the cells were fixed and the detection of BrdU incorporation was performed according to the manufacturers’ instructions. In short, the cells were incubated with anti-BrdU monoclonal antibody for 1 h, followed by application of goat anti-mouse IgG conjugated with peroxidase for 30 min at room temperature, protected from light. Thereafter, the cells were incubated with TMB peroxidase for 30 min while being protected from light and stopped by an acid stop solution. The optical density (OD) was measured at 450 nm using a spectrophotometer microplate reader (FLUOstar Omega; BMG Labtech, Ortenberg, Germany). Each experiment was performed independently in triplicate and carried out three times.

### 2.8. In Vitro Cell Scratch Assay

The effect of oxHA on the migration of corneal epithelial cells was determined using a scratch assay. For such, hTCEpi was seeded at a density of 9000 cells/well into 96-well plates and incubated overnight until cells were confluent. Cells were mechanically removed from the confluent monolayer by dragging a 10 μL (Neptune 191386) pipette tip linearly down the middle of the well. Standardization of the position and length of the scratch was accomplished using a guide placed at the bottom of each well. Any loose cells or debris were removed by washing twice with sterile PBS. Thereafter, 0.2% oxHA, HMWHA, LMWHA, ULMWHA (all prepared in sterile PBS), or PBS (vehicle control) in fresh media were added with a 1:10 dilution. The scratches were imaged from 0 to 30 h at 6 h intervals under an EVOS microscope. Finally, the tiff images were exported, and the wounded area was quantified using Image J 1.53k with an imageJ plugin [67]. The percentage of wounded area remaining was calculated using the formula (wounded area at 12 or 24 h/wounded area at 0 h) × 100. Each experiment was performed in triplicate and carried out twice.

### 2.9. In Vivo Wound Healing and Eye Drop Application

As previously described, 7–8-week-old *C57BL/6J* mice (male and female) were subjected to debridement wounds [68]. Before wounding, the mice were anesthetized by intraperitoneal injection of ketamine hydrochloride (80 mg/kg) and xylazine (10 mg/kg). Corneal epithelium was demarcated using a trephine (1.5 mm in diameter), and the epithelium within the demarcated area was removed using an AlgerBrush II (Alger Company, Inc., Lago Vista, TX, USA). Thereafter, the eye was washed with sterile PBS and a Polyvinyl Acetal eye spear in order to remove any loose cells. Fluorescein staining prepared by GloStrips (Amcon Laboratories, A01-33E, Amcon Laboratories) was used in order to ascertain that sharp wound edges were generated and that all cells within the demarcated area were successfully removed. The wounded area was determined by placing 1.5 μL of a 1 mg/mL fluorescein solution on the cornea, immediately after wounding and 12, 16, and 24 h after the injury. The ocular surface was imaged using a Zeiss Discovery V12 Stereo Microscope (Zeiss, Hebron, KY, USA). The wounded area was quantified by manually demarcating it as the region of interest by an investigator in a double-blind manner. For eye drop treatments, each eye was treated with 10 μL of PBS either containing 0.2% HA of different MWs or 0.2% oxHA every 15 min for the first 2 h and a single application 12 h and 16 h after injury. PBS only was used as a vehicle control. After the initial 2 h treatment, terramycin ointment was applied to the ocular surface, and the mouse was placed on a warming pad until it was awake enough to be placed in its home cage. Between 5 and 7 corneas were treated for each experimental point. All mice were bred and housed in a temperature-controlled facility with an automatic 12 h light/dark cycle at the Animal Facility of the University of Houston. All experimental procedures for handling the mice were previously approved by the Institutional Animal Care and Use Committee (IACUC) at the University of Houston under protocol 16-044. All animal care and use conformed to the ARVO Statement for the Use of Animals in Ophthalmic and Vision Research. 

### 2.10. Statistical Analysis

Data analysis was conducted using Microsoft Excel, Data-graph 4.6.1, and GraphPad Prism 9. Comparisons between multiple values were made using ANOVA followed by post hoc tests, and comparisons between the different treatments against vehicle control (PBS) or HMWHA were also carried out using paired *t*-test. *p* values < 0.05 were considered statistically significant. Unless stated otherwise, the error bars shown indicate SEM. Differences were considered as statistically significant when *p* < 0.05. 

## 3. Results

### 3.1. HA Fragmentation by H_2_O_2_ and HOCl

HMWHA of 2.67 × 10^6^ Da was incubated with H_2_O_2_ or HOCl for 4 h at different molar ratios in order to generate oxHA fragments of different MWs. The MW of the oxHA fragments was determined by both agarose gel electrophoresis and gel filtration HPLC. Smaller HA fragments were generated with increasing molar ratios of H_2_O_2_ or HOCl. Specifically, based on the agarose gel electrophoresis, incubating HMWHA with H_2_O_2_ at the molar ratio of 1:50 generated HA fragments that ranged from 2500 to 150 kDa, while at the molar ratio of 1:250, fragments of 250–100 kDa were generated (Figure 2A,B). Oxidizing HMWHA with HOCl at a ratio of 1:50 generated products ranging from 2500 to 250 kDa, a ratio of 1:250 generated products ranging from 2500 to 100 kDa, a ratio of 1:1250 generated products ranging from 300 to 50 kDa, and a ratio of 1:6250 generated products under 100 kDa (Figure 2A–C). Based on the analysis by gel filtration HPLC, H_2_O_2_ at a molar ratio of 1:50 led to a slight decrease in the MW of HA, which can be seen as a slight shift in the elution profile (Figure 2D). Although there is the generation of a small amount of oxHA of a lower MW after treatment with H_2_O_2_ at the molar ratio of 1:50, most of the HA remains as HMWHA. In contrast, when treating HMWHA with H_2_O_2_ at a molar ratio of 1:250, there is a significant shift in the elution profile of HA, with the peak shifting from ~13.52 min to ~14.16 min, and the majority of the HA elutes in the LMWHA range (Figure 2D). In the case of HOCl, treatment at a molar ratio of 1:50 did not yield a significant shift in the elution profile of HA, and the majority of the HA remains as HMWHA (Figure 2E). However, when HMWHA was treated with HOCl at a molar ratio of 1:250, there was a significant shift in the elution profile of HMWHA to mid-range MWHA, with the peak shifting from ~13.52 to ~14.00 min (Figure 2E). For 1:1250 and 1:6250, there was a significant shift in the elution profile of HMWHA to LMWHA and ULMWHA, respectively (Figure 2E).

### 3.2. Structural Characterization of HA following Oxidation by H_2_O_2_ and HOCl

For the 1H NMR spectrum of native HA, the peak at 2.0 ppm was assigned to the acetamido moiety of the N-acetyl-D-glucosamine residue. Following fragmentation reactions, HACl was obtained, as determined by the protein nuclear magnetic resonance (1H NMR) spectra, with evidence of proton peaks of saccharide rings (3.6–5.1 ppm) and a significant shift in CH3-(acetamido) protons from 2.0 to 2.3 ppm due to the formation of chloroamines (Figure 3C, asterisk). There is no clear distinction between HA and oxHA generated by H_2_O_2_ based on the NMR spectra, except for by the possible formation of aldehydes from 5.0 to 5.1 ppm (Figure 3A,B) due to divalent metal contaminants from HA preparations.

### 3.3. Effect of oxHA on the Viability of Corneal Epithelial Cells

The addition of HMWHA to the media of hTCEpi significantly promoted their viability by approximately 20% when compared to PBS alone, while LMWHA and ULMWHA had no effect (Figure 4A,B, respectively). Interestingly, the treatment of HMWHA with ROS, both H_2_O_2_ and HOCl, at all molar ratios except for HOCl at 1:6250 prevented HMWHA from promoting the viability of hTCEpi (Figure 4A). 

### 3.4. Effect of oxHA on the Proliferation of Corneal Epithelial Cells

The addition of HMWHA and LMWHA to the media of hTCEpi significantly promoted proliferation by approximately 20% compared to PBS alone (Figure 4B). Interestingly, the prior treatment of HMWHA with ROS, both H_2_O_2_ and HOCl, prevented it from promoting proliferation (Figure 4B). The treatment of hTCEpi with oxHA generated by treating HMWHA with HOCl at the molar ratios of 1:1250 and 1:6250, thereby generating oxHA products in the ULMWHA range containing chloramide groups, significantly inhibited proliferation when compared to PBS alone (Figure 4B). Curiously, native ULMWHA had no significant effect on the proliferation of hTCEpi (Figure 4B).

### 3.5. Effect of oxHA on the Migration of Corneal Epithelial Cells

When analyzing the effects of HA of different MWs on the migration of corneal epithelial cells, we found that LMWHA promoted the migration of hTCEpi, achieving statistical significance at 12 h, while HMWHA and ULMWHA had no significant effect on the migration of hTCEpi (Figure 5A). Interestingly, oxHA generated by treating HMWHA with H_2_O_2_ at the molar ratio of 1:250, thereby generating oxHA products in the LMWHA range, also promoted the migration of hTCEpi, achieving statistical significance at 12 h (Figure 5A). In contrast, oxHA generated by treating HMWHA with H_2_O_2_ at the molar ratio of 1:50, which has a very subtle effect on the overall MW of HA, had no significant effect on the migration of hTCEpis when compared to that of HMWHA and PBS alone (Figure 5A,B). oxHA generated by treating HMWHA with HOCl at the molar ratios of 1:50 and 1:250, thereby generating oxHA products that are smaller than HMWHA but do not reach the LMWHA range or contain chloramide groups, promoted the migration of hTCEpi, but not to the same extent as LMWHA, and the values did not reach statistical significance (Figure 5B,C).

### 3.6. Effect of oxHA on Corneal Epithelial Wound Healing In Vivo

The effects of oxHA on corneal epithelial wound healing were also investigated and compared to those of native HA of different MWs in vivo (Figure 6). HMWHA, MMWHA, and LMWHA promoted corneal epithelial wound healing when compared to PBS alone, reaching significance at the 24 h time point (Figure 6A,B). The oxHA fragments generated by H_2_O_2_ at a molar ratio of 1:50 also promoted corneal epithelial wound healing, similar to HMWHA, but did not reach statistical significance when compared to PBS alone (Figure 6C,D). In contrast, previously oxidizing HMWHA with H_2_O_2_ at a molar ratio of 1:250 ablated the capability of HMWHA to promote corneal epithelial wound healing (Figure 6C,D). The oxHA generated by HOCl promoted corneal epithelial migration when compared to PBS alone, but did not reach statistical significance (Figure 6E,F). Therefore, the oxHA fragments generated by oxidizing HMWHA with HOCl also promoted corneal epithelial wound healing (Figure 6E,F).

## 4. Discussion

This study aimed to investigate whether fragments produced after oxidizing HA present distinct physiological properties when compared to size-matched native HA. Although studies have explored the effects of different ROS on HA, including characterizing the structure of a variety of oxHA products, most studies have focused on the role HA has in protecting tissues from ROS and not on the potential physiological functions of the oxHA fragments that are produced. Studies have demonstrated nicely that HA is highly susceptible to oxidative stress, and, as such, can serve as a “sponge” in tissues by undergoing chemical modifications by ROS in lieu of other molecules [20,21,22,23,24,25,45,46,47,48]. However, to the best of our knowledge, this is the first study to investigate whether chemically modified oxHA fragments produced by ROS present altered physiological properties, when compared to native HA of the same MW. Firstly, we characterized the MW and structure of the oxHA fragments generated by H_2_O_2_ and HOCl. Subsequently, we compared the physiological properties of oxHA compared to size-matched native HA, both in vitro and in vivo. Our study clearly shows that oxidized fragments produced after the oxidation of HMWHA have altered physiological functions when compared to native HMWHA. Importantly, our data demonstrate that the altered physiological properties of oxHA are caused both by a change in the MW of HA and by a change in the chemical structure of the oxHA fragments.

Studies have shown that the molecular weight of HA dictates its physiological functions [38,39], which is attributed, at least in part, to the fact that the length of the HA chain dictates how it binds and interacts with HA-specific cell surface receptors, leading to either the activation or deactivation of specific signaling pathways [38]. For instance, HMWHA can lead to the clustering of the extracellular domain of CD44 and stimulate Hippo signaling, whereas LMWHA is unable to promote the clustering of CD44 and instead leads to inhibition of Hippo signaling [34]. Our study clearly demonstrates that HMWHA exposed to ROS leads to the fragmentation of the HA chain, generating oxHA fragments that range from less than 100 kDa to over 2000 kDa. Importantly, this fragmentation of the HA chain leads to changes in its physiological properties. Previous studies have demonstrated that HA subjected to oxidation by certain ROS can generate oxHA fragments with modified chemical structures at the non-reducing end [49]. HOCl has been shown to generate HA fragments with chloramides and amidyl radicals [21,22,23,24,25]. Therefore, HA not only undergoes fragmentation by HOCl, but also undergoes changes to its chemical structure. In this study, H_2_O_2_ and HOCl were selected as ROS since they have both been shown to fragment HA, with HOCl altering the chemical structure of HA [21,22,23,24,25]. In contrast, H_2_O_2_ in the absence of ferrous cations, such as cupric ions, does not generate hydroxy radicals, which are essential to initiate the radical attack to induce intermediate HA species, therefore, not altering the chemical structure of the HA fragments [50,69]. Our study confirmed that the oxHA produced by HOCl undergoes a chemical modification with the introduction of chloramide groups, while the oxHA produced by H_2_O_2_ does not. Interestingly, our study revealed that oxHA in the ULMWHA range containing chloramide groups has distinct physiological functions when compared to size-matched native ULMWHA. Specifically, we found that at the concentration of 0.2 mg/mL, HMWHA and LMWHA significantly promoted hTCEpi proliferation, whereas the oxHA produced by H_2_O_2_ prevented HA from inhibiting proliferation, likely caused by the breakdown of the HMWHA into fragments. In contrast, oxHA produced by HOCl in the 1:1250–1:6250 range inhibited proliferation, which is different to the effects of native LMWHA and ULMWHA. Thus, oxHA produced by HOCl had distinct effects on corneal epithelial cell proliferation when compared to size-matched native HA, which we can infer is caused by the altered chemical structure of oxHA. Additionally, the oxHA produced by H_2_O_2_ had a similar effect on corneal epithelial cell migration as size-matched native HA, while the oxHA produced by HOCl had a distinct effect on corneal epithelial cell migration when compared to size-matched native HA. Our data demonstrate a need for future research on whether oxidized forms of HA with altered chemical structures bind differently to HA surface receptors and/or have distinct interactions with HA binding proteins within the ECM when compared to size-matched native HA.

HA is a key ECM component in various stem cell niches, including bone marrow mesenchymal stem cells [70], umbilical cord mesenchymal stem cells [71], and LESCs [72]. We speculate that in tissues, HA forms an optimal microenvironment for supporting stem cells, as well as serving as a means of protecting them from ROS challenges. This is supported by the fact that corneal epithelial progenitor cells, which are surrounded by an HA-rich niche, have been shown to be resistant to H_2_O_2_ damage [73]. In fact, HA matrices have shown prominent antioxidant characteristics in various tissues [40,41,42,43]. It has been suggested that in tissues, the protective effect exerted by HA against oxidative damage comes mainly from HA undergoing chemical modifications by ROS in lieu of other molecules, and not by altering the production of ROS and/or the synthesis of antioxidant enzymes [74]. Following corneal injuries, including corneal debridement wounds, there is an excess of ROS produced, and HA would therefore be prone to ROS-mediated fragmentation [14]. Additionally, we have previously shown that HA is upregulated the peripheral cornea and limbal region after an epithelial injury and is upregulated throughout the limbal epithelium and corneal epithelium and in the stroma after an alkali burn [30,33,35]. Therefore, it is important to understand the physiological functions of oxHA and whether oxHA has an effect on corneal wound healing that is distinct from native HA.

Previous studies have shown divergent results regarding the effects of different forms of native HA on cell migration and wound healing [39,75,76,77]. Phillips AO et al. have shown that HMWHA promotes cell migration and wound healing in cells such as proximal tubular cells [76,77], while others have shown that HA of a lower MW promotes migration and wound healing [39,75]. Notably, the physiological activity of HA has been shown to occur in a concentration-dependent manner. For example, epithelial migration increased proportionally to the HA (800–1400 kDa) concentration, until reaching a plateau at 2 mg/mL [78]. HA eye drops at concentrations of 2 mg/mL and 4 mg/mL HA (800–1400 kDa) promote re-epithelialization after debridement wounds in rabbits [78]. The daily application of 1 mg/mL and 2.5 mg/mL HA (873 kDa) promotes rabbit corneal epithelium migration in vivo after debridement wounds [79]. Eye drops containing HA at a concentration of 1.8 mg/mL promote corneal wound healing in diabetic patients after corneal epithelial removal in pars plana vitrectomy [80]. Additionally, HA eye drops at a concentration of 1 mg/mL significantly promote corneal epithelial cell migration in rabbit corneas with HA ranging from 90 kDa to 2800 kDa having similar effects, indicating that at certain concentrations, HA promotes corneal wound healing irrespective of size [81]. Overall, previous studies have used HA eye drops at concentrations mainly ranging from 1 mg/ mL to 4 mg/mL [78,81], but studies are still required to determine the physiological levels of HA in the corneal epithelium, corneal stroma, and tear film before the optimal concentration of HA can be determined for in vitro and in vivo corneal wound healing assays [82]. In our study, we used HA at a concentration of 2 mg/mL for the in vivo experiments and found that all forms of native HA, with the exception of ULMWHA, significantly promoted wound healing. Overall, to date, evidence suggests that native HA promotes corneal wound healing in both a size- and concentration-dependent manner [78,79]. In addition to this, our data suggest that HA with altered chemical structures at the non-reducing end has physiological functions distinct from those of native HA. 

To the best of our knowledge, no studies to date have investigated whether the change in the chemical structure of HA following ROS affects its physiological functions. Therefore, our study comparing oxHA products with and without chemical modifications compared to size-matched HA standards enabled us to verify whether altering the chemical structure of HA alters its physiological properties and how ROS can affect corneal wound healing by cleaving HA. We found that oxHA produced by HOCl did present some distinct physiological effects when compared to size-matched native HA, which can be attributed to the altered chemical structure of HA. Taken together, this study shows that HMWHA is fragmented into HMWHA, MMWHA, LMWHA, and ULMWHA by ROS in a concentration-dependent manner. Moreover, the physiological effects of HA are altered after chemical modifications by ROS. This is of particular relevance when using HA-based therapies in ocular pathologies with elevated levels of ROS. Future research should be dedicated to investigating the interactions of oxHA with the various cell surface receptors and hyaldherins when compared to native HA. Such research would be invaluable in determining how oxHA affects cellular processes differently to native HA.

## 5. Conclusions

HA is highly susceptible to ROS in a concentration-dependent manner, producing oxHA products of reduced MWs with or without altered chemical structures. Both the reduced MW and the altered chemical structure of the oxHA products affect the physiological properties of HA.

## Figures and Tables

**Figure 1 biomolecules-13-01385-f001:**
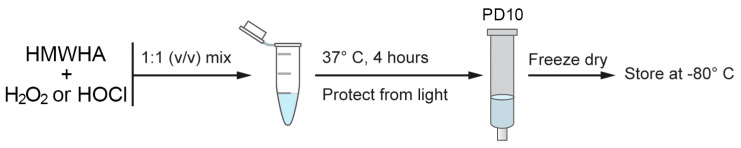
Schematic representation of the preparation of the oxidized forms of HA. Both H_2_O_2_ and HOCl were used to oxidize HA at different molar ratios. Samples were incubated at 37 °C for 4 h, and thereafter, H_2_O_2_ and HOCl were removed from the samples by gel filtration chromatography using a PD10 desalting column. The samples were quantified, aliquoted, and stored for use.

**Figure 2 biomolecules-13-01385-f002:**
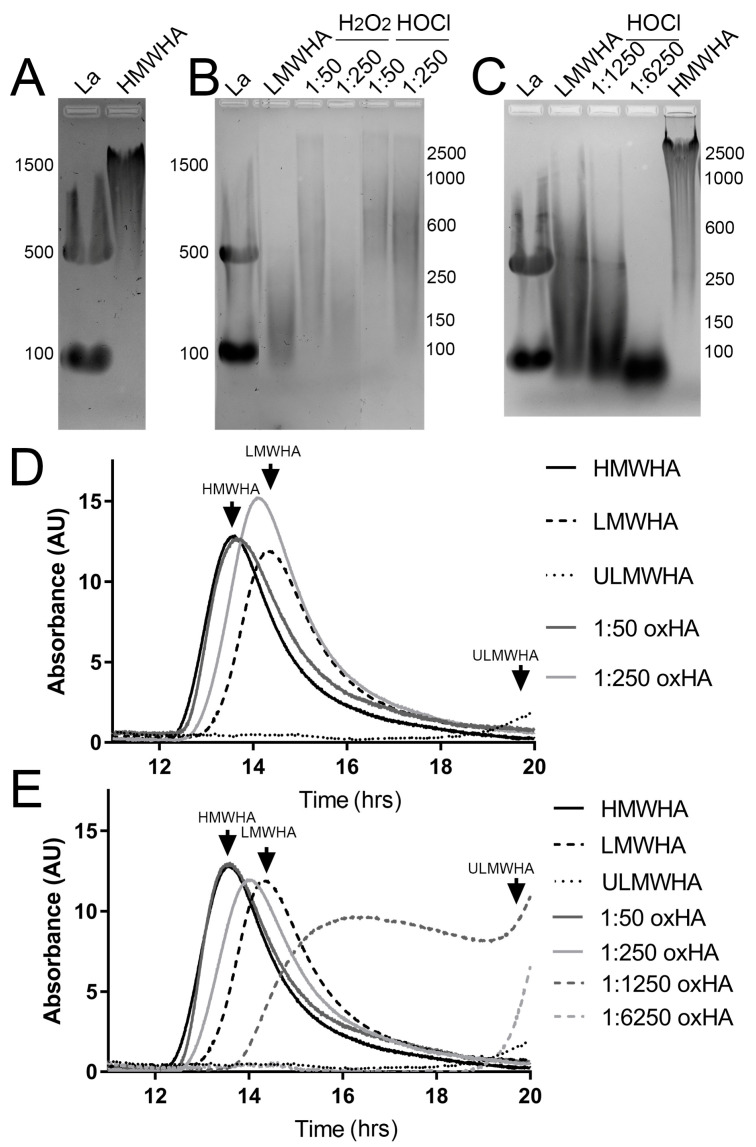
Characterization of the molecular weight of oxHA. The molecular weight of the differently prepared oxHA samples was determined by agarose gel electrophoresis (**A**–**C**) and by gel filtration using HPLC (**D**,**E**). (**A**) Ladder (La) and HMWHA were subjected to agarose gel electrophoresis. (**B**) La, HMWHA, and both H_2_O_2_ and HOCl at molar ratios of 1:50 and 1:250 were subjected to agarose gel electrophoresis. (**C**) La, LMWHA, HMWHA, and HOCl-treated HMWHA at the molar ratios of 1:1250 and 1:6250 were subjected to agarose gel electrophoresis. (**D**) HMWHA, LMWHA, ULMWHA, and H_2_O_2_-treated HMWHA were subjected to gel filtration using HPLC. (**E**) HMWHA, LMWHA, ULMWHA, and HOCl-treated HMWHA were subjected to gel filtration using HPLC.

**Figure 3 biomolecules-13-01385-f003:**
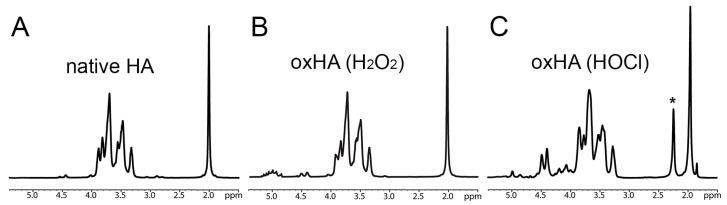
Chemical characterization of oxHA by NMR. ULMWHA, oxHA in the LMWHA range generated by H_2_O_2_ oxidation and oxHA in the ULMWHA range generated by HOCl oxidation were analyzed. (**A**) ^1^H NMR spectra of native HA. (**B**) ^1^H NMR spectra of oxHA produced by H_2_O_2_. (**C**) ^1^H NMR spectra of oxHA produced by HOCl are presented. The asterisk shows the chemical shift in acetamido protons upon chlorination.

**Figure 4 biomolecules-13-01385-f004:**
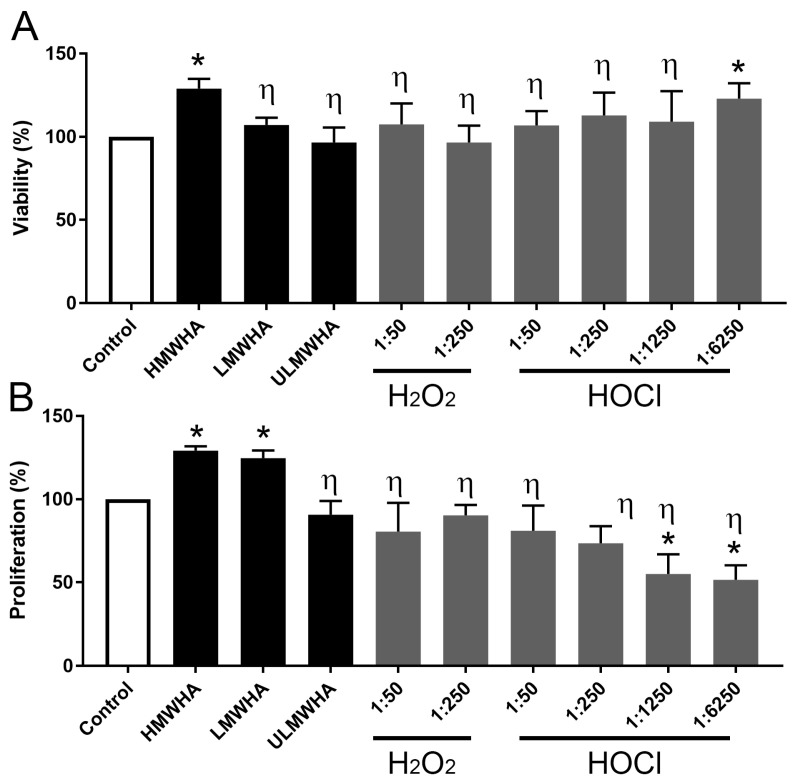
Biological effects of different oxidized forms of HA on human corneal epithelial cell viability and proliferation. (**A**) hTCEpi were treated with HMWHA, LMWHA, ULMWHA, and both H_2_O_2_- and HOCl-treated HMWHA, and viability was assayed using the CCK-8 kit. Values are presented as percentage change compared to the vehicle control (PBS). (**B**) hTCEpi was treated with HMWHA, LMWHA, ULMWHA, and both H_2_O_2_- and HOCl-treated HMWHA, and proliferation was assayed using the BrdU assay kit. Values are presented as percentage change compared to the vehicle control (PBS). η represents *p* ≤ 0.05 in reference to HMWHA and * represents *p* ≤ 0.05 in reference to PBS control. Each experiment was carried out in triplicate and carried out three times.

**Figure 5 biomolecules-13-01385-f005:**
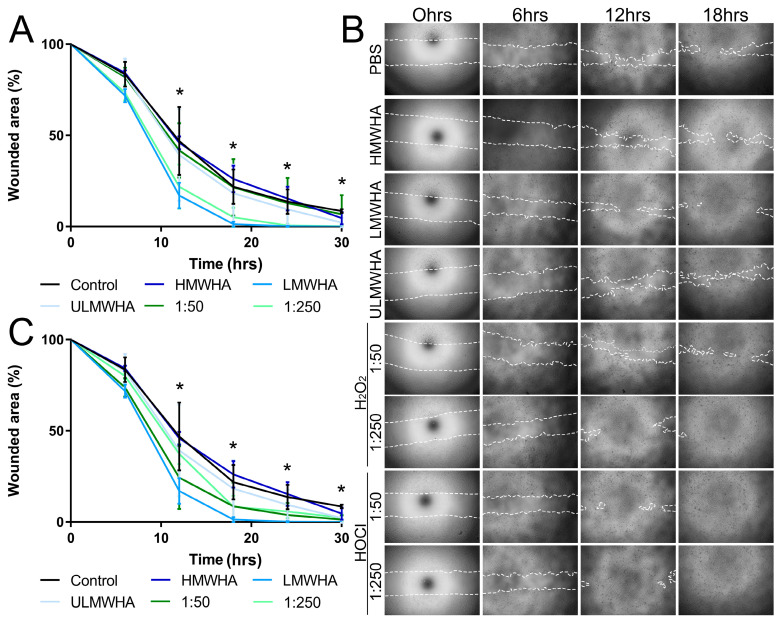
Biological effects of different oxidized forms of HA on corneal epithelial cell migration in vitro. (**A**) Confluent hTCEpi were subjected to scratch wounds and maintained in media containing 0.02 mg/mL HMWHA, LMWHA, ULMWHA, or H_2_O_2_-treated HMWHA. Images were collected 0, 6, 12, 18, 24, and 30 h after injury, the wounded area was calculated using Image J, and data are presented as the percentage of wounded area remaining compared to 0 h. (**B**) Representative images of the wounded area over time. The boundaries of the wounded area are demarcated with a dashed white line. (**C**) Confluent hTCEpi were subjected to scratch wounds and maintained in media containing 0.02 mg/mL HMWHA, LMWHA, ULMWHA, or HOCl-treated HMWHA. Images were collected 0, 6, 12, 18, 24, and 30 h after injury, the wounded area was calculated using Image J, and data are presented as the percentage of wounded area remaining compared to 0 h. Each experiment was carried out in triplicate and carried out three times. * represents *p* ≤ 0.05 in reference to PBS control.

**Figure 6 biomolecules-13-01385-f006:**
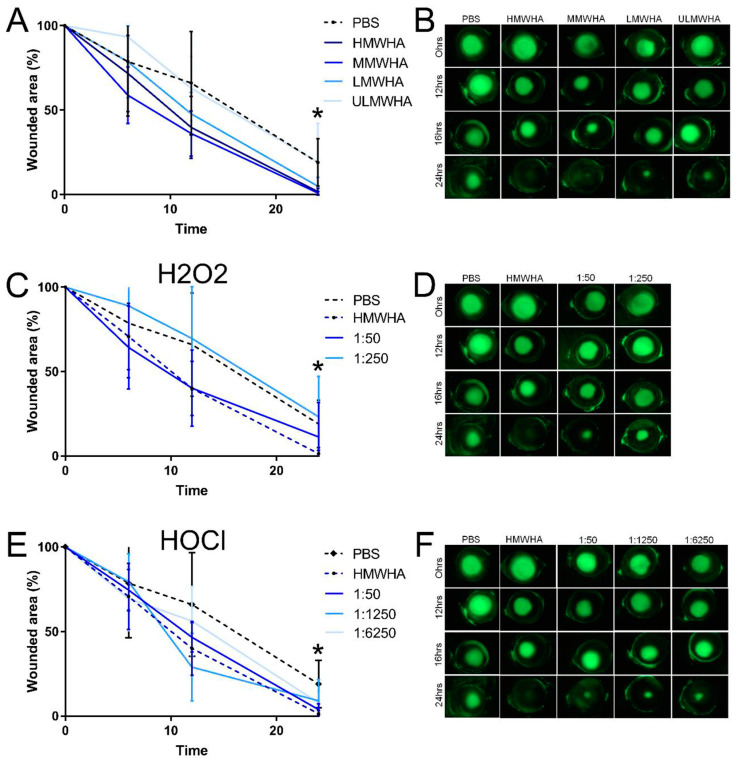
Biological effects of different oxidized forms of HA on corneal epithelial cell migration in vivo. C57BL6 mice were subjected to debridement wounds and topically treated with HMWHA, LMWHA, ULMWHA, or oxHA generated by H_2_O_2_ or HOCl at 2 mg/mL. (**A**) Debridement wounds were treated with HA of different MWs and compared to PBS. The wounded area was evidenced with fluorescein, the wounded area was calculated with imageJ, and data are presented as the percentage of wounded area remaining compared to the wounded area at 0 h. (**B**) Representative images of the data are presented in panel (**A**). (**C**) Debridement wounds were treated with oxHA generated by H_2_O_2_, compared to PBS and HMWHA. The wounded area was evidenced with fluorescein, the wounded area was calculated with imageJ, and data are presented as the percentage of wounded area remaining compared to the wounded area at 0 h. (**D**) Representative images of the data presented in panel (**C**). (**E**) Debridement wounds were treated with oxHA generated by HOCl, compared to PBS and HMWHA. The wounded area was evidenced with fluorescein, the wounded area was calculated with imageJ, and data are presented as the percentage of wounded area remaining compared to the wounded area at 0 h. (**F**) Representative images of the data are presented in panel (**E**). * represents *p* ≤ 0.05. Images for PBS and HMWHA in panel B, D, and F are adapted from [64], and copyright has been gained from the authors and the journal TVST.

## Data Availability

No archived datasets were analyzed or generated during this study and all the data that were created have been presented in their entirety.

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
