# Peer review of "ROS-Mediated Fragmentation Alters the Effects of Hyaluronan on Corneal Epithelial Wound Healing"

_biomolecules, 2023, doi:10.3390/biom13091385_

Round 1

Reviewer 1 Report

The manuscript “ROS mediated fragmentation alters the effects of hyaluronan on corneal epithelial wound healing” investigates the effect of HOCl and H2O2 exposure on hyaluronan and the effect of oxidized hyaluronan on corneal epithelial cell viability, proliferation, and wound healing. The introduction is well-written, and the methods include sufficient details. It is an interesting manuscript with novel results. The following suggestions may be addressed:

1.      In Fig 2, HA bands are hard to distinguish, and the loading amount seems low.

2.      The data for cell viability and proliferation can be presented as percentages rather than AUC.

3.      For Fig 4 A, HOCl 1:6250 bar does not seem to be statistically different from HMWHA. 

4.      Only viability data is presented for mouse corneal epithelial cells; the data’s outcome is similar to human corneal epithelial cells and does not add much additional value. This data may be removed.

5.      The shades of colors for the different curves in Fig 5 are hard to distinguish. The graph has no statistical significance. Is the data not statistically significant?

6.       Images for the scratch assay are of poor quality.

7.      The discussion can be made more concise with emphasis on the biological basis of a differential effect of various OXO-HAs.

Reviewer 2 Report

The manuscript by Lin et al characterizes the effect of different molecular weights of hyaluronic acid, its oxidized form and effect on corneal epithelium in vitro and in vivo. While largely well done, the inferences drawn from the study are over interpreted considering the experimental design implemented.

1) In the methods, the authors only use HWMHA to generate oxidized form of the HA. In the introduction they acknowledge that HMW and LMW HA may have differences and thus impart differential behaviors on cell fate and responses. Yet, the choice to only use HMWHA for this critical element is surprising. 

2) The rationale for the utility of a hypochlorite to oxidize HA and how this corresponds to the nature of oxidized HA in vivo is unclear. While the authors quantify using NMR and HPLC the types of HA in vitro, It is surprising that the authors choose to not characterize the type of HA in corneal tissues before and after an injury. Further, this is a missed opportunity to identify the type of oxidized HA (if any) after a corneal injury and during remodeling considering this is the primary motivation for the study.

3) In the methods the authors state HMWHA was used for oxidation, yet in Fig 3, they represent native ULMWHA and the fragments produced. It is confusing as represented which HA was used for what experiment for oxidation

4) Statistics: The study is riddled with incorrectly or unadequately described/performed statistics. In the methods the authors simply state ANOVA followed by Holm-Bonferroni correction was used suggesting a one-way ANOVA. Fig. 4, it appears the authors use one-way ANOVA. When in fact they are testing the effect of HA fragments derived from two oxidizers on (an unknown type of HA) and different HA on cell viability compared with untreated cells. Since this suggests multiple variables (oxidization and type of HA) an one-way ANOVA is inappropriate. Thus the significant effects seen on viability (Fig. 4a) by 1:6250 HOCl on HMWHA/control are likely spurious.

5) Fig.5: the authors indicate in methods a scratch wound was performed. Yet scratches and separation of cells after a scratch in Fig 5b and D are not seen and appear to be concentric. Thus inference of the data is unclear. Statistics: No statistical testing is performed to assess differences here. 

6) Impact on wound healing in vivo. This is a spuriously forced result. The authors represent the data from diff oxidation of HMWHA over time. This is confusing since the results and discussion are comparing across the different types of oxidation and they talk about implication of oxHA on wound healing compared with control. The authors did not do that here. They choose to compare wihtin same group with time point 0. 

7) It is unclear if oxidation of all MW of HA are comparable considering different polymers (bio and synthetic) oxidize differently at different molecular weight. As such, the biological effects may also differ.

8) The authors in discussion do finally explain that post oxidation, fragmentation of HA results in HA of various MW. This is not clear in the intro, methods or results. Perhaps this needs to be cleared up front. Also, the authors loftily claim physiologcal effects are seen by different ROS. This is untrue as such a comparison is not performed in this study. If anything, the changes the authors are seeing (not apparent to this reviewer) is an error of the statistical method they are using. 

The manuscript is written in a very convoluted manner. Simplifying the experimental design with a schematic/flowchart may help

Reviewer 3 Report

The papers is fit for the journal scope and it is really well written. However few considerations need to be taken into account:

-Line 85: please, specify the acronymous HOCl;

-Section 2.8, lines 250-251: in my opinion it is important to introduce the formula used to determine the percentage of healing over time;

-Figure 5 (B and D): provide a better quality figure. 

Minor spelling check. 

Reviewer 4 Report

1. Some of grammar mistakes were found out, and it would be better if the authors revise the whole article and polish the language. For example,

Page1 line42 “Unbalanced oxidative chemical reactions that buildup excess ROS leads to oxidative stress”.

2. The number of the Discussion section should be "4." instead of "3.".

3. It would better to label the x-axis in Figure 2, 4, and 5 as "Time (hrs)" to indicate the unit of measurement as hours.

4. Background descriptions for wound healing can be strengthened by citing 10.1016/j.cej.2023.141852; 10.1021/acsami.1c25014 and what are the advantages of the current work compared to published articles?

5. The forms "CCK8" and "CCK-8" mentioned in the article should be standardized, and the authors should consistently use one form throughout. Additionally, it is recommended that the authors clarify the abbreviation for "Cell Counting Kit-8" as "CCK-8" after its first mention.

6. The format of the captions below the figures is not consistent, and it is recommended to ensure uniformity throughout.

7. The bands appear blurry in Figure 2. Therefore, I would recommend the authors to consider repeating the agarose gel electrophoresis step.

8. The contrast in the black and white images may not be sufficient, resulting in less clarity. I would suggest replacing the images of different time periods of the wound in Figure 5B and Figure 5D with color images.

9. The study did not investigate the effects of oxHA fragments on other cellular processes, such as differentiation or angiogenesis, which were mentioned in the “Introduction” section. Future studies could explore the effects of oxHA fragments on these processes.

Round 2

Reviewer 1 Report

The authors have adequately addressed all the suggestions.